# Association of lipocalin-2 and low-density lipoprotein receptor-related protein-1 (LRP1) with biomarkers of environmental enteric dysfunction (EED) among under 2 children in Bangladesh: results from a community-based intervention study

Md. Mehedi Hasan,[1] Md. Amran Gazi [ID],[1] Subhasish Das,[1] Shah Mohammad Fahim,[1] Farzana Hossaini,[1] Md. Ashraful Alam,[1] Mustafa Mahfuz,[1] Tahmeed Ahmed [ID] [1,2]

For numbered affiliations see end of article.

**Correspondence to**
Md. Amran Gazi; amran.gazi@icddrb.org

## ABSTRACT

**Background** Environmental enteric dysfunction (EED) is thought to occur from persistent intestinal inflammation. Studies also revealed the association of lipocalin-2 (LCN2) and low-density lipoprotein receptor-related protein-1 (LRP1) with intestinal inflammation. Therefore, we intended to explore the relationship of LCN2 and LRP1 with gut inflammation and biomarkers of EED in Bangladeshi malnourished children.

**Methods** A total of 222 children (length-for-age z-score (LAZ) <−1) aged 12–18 months were enrolled in this study in a cross-sectional manner. Among the participants, 115 were stunted (LAZ <−2) and 107 were at risk of being stunted (LAZ −1 to −2) children. Plasma and faecal biomarkers were measured using ELISA. Spearman's rank correlation was done to see the correlation among LCN2, LRP1 and biological biomarkers.

**Results** LCN2 correlates positively with myeloperoxidase (r=0.19, p=0.005), neopterin (r=0.20, p=0.004), calprotectin (r=0.3, p=0.0001), Reg1B (r=0.20, p=0.003) and EED score (r=0.20, p=0.003). Whereas, LRP1 correlates negatively with myeloperoxidase (r = −0.18, p=0.006), neopterin (r = −0.30, p=0.0001), alpha-1-antitrypsin (r = −0.18, p=0.006), Reg1B (r=−0.2, p=0.003) and EED score (r = −0.29, p=0.0001).

**Conclusions** Our findings imply that LCN2 might be a promising biomarker to predict gut inflammation and EED. Whereas, increased level of LRP1 may contribute to alleviating intestinal inflammation.

## INTRODUCTION

Environmental enteric dysfunction (EED) is a subacute intestinal inflammation highly prevalent in low-income and middle-income countries, which is defined by crypt hyperplasia, villous atrophy, reduction of mucus layer and infiltration of lymphocytes in

### What is known about the subject?

► Environmental enteric dysfunction (EED) is highly prevalent in low-income and middle-income countries, which affects approximately 40% of all children with age less than 5 years.
► EED causes linear growth faltering or stunting which has the manifestations including reduced neurodevelopmental and cognitive function and increased morbidity and mortality in childhood.
► The main challenge to tackle the EED is lack of non-invasive biomarker signature to diagnose the EED and take proper treatment initiative.

### What this study add?

► There are significant positive correlation of lipocalin-2 and significant negative correlation of low-density lipoprotein receptor-related protein-1 (LRP1) with biomarkers of EED.
► Study findings strongly infer that lipocalin-2 might be a promising biomarker to predict gut inflammation and EED.
► Study findings also imply that LRP1 could be a therapeutic target to alleviate intestinal inflammation.

lamina propria.[1] EED affects approximately 40% of all children with age less than 5 years in resource poor settings and causes linear growth faltering or stunting.[2] It is thought to occur from repeated enteric infections due to poor hygiene and unsanitary environmental conditions, which in turn results in intestinal inflammation, a major driver of occurring EED.[3] Studies have elicited the relation of

lipocalin-2 (LCN2) and low-density lipoprotein receptor-related protein-1(LRP1) with intestinal inflammation. LRP1 knockout has been found to be associated with increased intestinal inflammation in animal model,[4] and LCN2-deficient mice are more prone to bacterial infection.[5] On the other hand, higher expression of LCN2 has been reported in various inflammatory conditions.[6] Hence, it is necessary to explore the relation of LCN2 and LRP1 with intestinal inflammation in the context of EED.

LCN2 is a bacteriostatic peptide, a member of lipocalin super family, known as neutrophil gelatinase B-associated lipocalin.[7 8] One of the most recognised functions of LCN2 is to prevent the bacterial uptake of iron crucial for their growth.[9 10] It has been reported that LCN2-deficient mice show higher mortality rates following infection with *Escherichia coli* than wild type.[5] Study also reported that LCN2 shows preventive mechanism against intestinal inflammation through bacterial clearance by phagocytosis in macrophages.[7] Moreover, it exerts defensive mechanism against *Salmonella typhimurium, Klebsiella pneumoniae* and *E. coli* in different studies.[11–13] Besides the defensive mechanism, LCN2 has emerged as promising biomarker of inflammation. It has been reported as hallmark for the onset of kidney diseases as well as indicator of inflammation in various diseases.[6 14 15] However, there is no study that has conducted to investigate the level of LCN2 in Bangladeshi malnourished children. Therefore, the relation of LCN2 with intestinal inflammation and EED are yet to be elucidated.

LRP1 is an endocytic receptor that facilitates the binding and endocytic transportation of a broad range of biological ligands as well as various molecules function in immune system.[16–18] It has a vital role in phagocytosis which is an essential component of the innate immune response, crucial for removal of infectious agents.[16 19] Moreover, LRP1 promotes tissue-resident macrophage-survival by activating Akt pathway.[20] Macrophages show the initial defensive mechanism by eliminate noxious agents including bacteria, irritants and necrotic cells,[21] whereas LRP1 deficiency causes increased macrophage apoptosis.[22] LRP1 also mediates pathogen elimination by scavenging membrane lipoproteins of microorganism during the infection.[23] Therefore, it can be hypothesised that LRP1 may reduce the gut inflammation by showing defensive mechanism against microbial invasion. In this study, we intended to investigate the level of LCN2 and LRP1 in malnourished children of Bangladesh and their relations with faecal markers of intestinal inflammation and EED.

## METHODS

### Study participants, study site and ethical consideration

To perform the analysis for this study, data were taken from 'Bangladesh environmental enteric dysfunction' study. The study protocol was published earlier.[24] It is a community-based intervention study to validate non-invasive biomarkers of EED. A total of 222 participants were included in this analysis in a cross-sectional manner, among them 115 were stunted (length-for age z-score (LAZ) <-2) and 107 were at risk of being stunted (LAZ −1 to −2). Study participants were enrolled from Bauniabadh, a slum of Mirpur, Dhaka, Bangladesh. The study protocol (PR-16007) was approved by the institutional review board of International Centre for Diarrhoeal Disease Research, Bangladesh (icddr,b). Written consent was obtained from mother of the participant after explaining about the study.

### Data acquisition, sample collection and laboratory assay

Anthropometric and socioeconomic data were collected from study participants by trained staff. LAZ, weight-for length z-score (WLZ), weight-for age z-score (WAZ), mid-upper arm circumference (MUAC) and head circumference of the study participants were calculated as anthropometric indices. A total of 4 mL of blood was collected by medical doctor following all the aseptic conditions and transported to laboratory maintaining cold condition (−4°C). Plasma was collected by centrifugation of blood at 3000 ×g for 10 min. Aliquots of biological samples were stored at −80°C until biomarker analyses. Plasma biomarkers including LRP1 (Biomatik, USA), sCD14 (R& D system, USA), C-reactive protein (CRP) (Immundiagnostik, Germany), alpha-1 acid glycoprotein (Alpco, NH) and ferritin (Orgentec, Germany) were measured by using available ELISA kits. Atomic absorption spectrometry was used to measure the level of zinc in plasma. Stool sample were used to detect the level of biomarkers including LCN2 (R&D system, USA), neopterin (NEO) (GenWay Biotech, California, USA), myeloperoxidase (MPO) (Alpco, New Hampshire, USA), alpha-1 antitrypsin (A1AT) (Biovendor, North Carolina, USA), Reg1B (TechLab, USA), calprotectin (BUHLMANN, Switzerland) by using available ELISA kits. All the laboratory analyses were performed at the parasitology laboratory of icddr,b.

### Statistical analyses

Data analyses were performed to present the characteristics of the study participants. Data were presented as median with IQRs when variables show skewed distributions, while categorical variables were showed as frequencies with percentages. The variables showing normal distribution were presented as mean with SD. Mann-Whitney U test was used to detect the differences in the age, family income, plasma biomarkers including CRP, alpha-1-acid glycoprotein (AGP), sCD14, LRP1, ferritin and zinc as well as faecal biomarkers including MPO, NEO, A1AT, calprotectin, Reg1B and LCN2; whereas $\chi^2$ test was performed for comparing categorical variables of the participants. Student's t-test was performed to see the differences in length, weight, MUAC, head circumference, LAZ, WLZ, WAZ, maternal height and WAMI index (Water, sanitation, hygiene,

Asset, Maternal education and Income index, ranging from 0 to 1).[25] Spearman's correlation coefficient (r) was calculated to see the correlation of LRP1 and LCN2 with other faecal biomarkers and EED score (ranging from 0 to 10) that has been calculated from MPO, NEO and A1AT using the method published earlier.[26] In this analysis, $p<0.05$ was considered statistically significant. All the analyses were performed using STATA V.13.

## Patient and public involvement

No patient or public were involved with the research design, protocol development and participant's enrolment. None of them will be involved during dissemination of findings.

## RESULTS

### Anthropometric, socioeconomic and biochemical parameters of the study participants

Table 1 presents the anthropometric and socioeconomic characteristics of the participants. Among the participants nearly half were female.

The levels of plasma biomarkers including LRP1, AGP, CRP, sCD14, ferritin and zinc and faecal biomarkers including MPO, NEO, A1AT, calprotectin, Reg1B and LCN2 were presented in table 2.

### Correlation of LCN2 and LRP1 with biomarkers and EED score

A significant positive correlation was found among LCN2 and different biomarkers of intestinal inflammation like MPO, NEO, calprotectin and Reg1B. LCN2 also positively correlated with EED score. However, LRP1 was negatively and significantly correlated with MPO, NEO, A1AT,

**Table 1** Anthropometric and socioeconomic characteristics of the study participants

| | Overall (n=222) |
|---|---|
| Child age (month), median (Q1, Q2) | 18.3 (16.5, 20.3) |
| Female sex*, n (%) | 116 (52.3) |
| Weight in kg, mean (SD) | 9.05 (0.96) |
| Length in cm, mean (SD) | 76.7 (2.85) |
| MUAC in cm, mean (SD) | 14.2 (0.93) |
| Head circumference in cm, mean (SD) | 44.7 (1.38) |
| Weight for age z score, mean (SD) | −1.44 (0.86) |
| Length for age z score, mean (SD) | −1.82 (0.79) |
| Weight for length z score, mean (SD) | −0.774 (0.91) |
| Maternal height in cm, mean (SD) | 149 (5.09) |
| Family income per month (US$),median (Q1, Q2) | 179 (124, 239) |
| Improved sanitation*, n (%) | 152 (68.5) |
| WAMI†, mean (SD) | 0.598 (0.134) |

\* Categorical variables (sex and improved sanitation).
† WAMI: Water, sanitation, hygiene, Asset, Maternal education and Income index (ranging from 0 to 1).
MUAC, mid-upper arm circumference.

**Table 2** Levels of plasma and faecal biomarkers of the study participants

| | Overall (n=222) |
|---|---|
| **Plasma biomarkers** | |
| LRP1 (ng/mL), median (Q1, Q3) | 1080 (796, 1390) |
| AGP (mg/dL), median (Q1, Q3) | 91.9 (69.0, 118) |
| CRP (mg/L), median (Q1, Q3) | 0.820 (0.284, 2.84) |
| sCD14 (ng/mL), median (Q1, Q3) | 1760 (1470, 2120) |
| Ferritin (ng/mL), median (Q1, Q3) | 13.8 (7.17, 25.1) |
| Zinc (mg/L), median (Q1, Q3) | 0.760 (0.68, 0.84) |
| **Faecal biomarkers** | |
| MPO (ng/mL), median (Q1, Q3) | 1720 (796, 3720) |
| Neopterin (nmol/L), median (Q1, Q3) | 1240 (567, 2690) |
| A1AT (mg/g), median (Q1, Q3) | 0.294 (0.13, 0.52) |
| Calprotectin (μg/g), median (Q1, Q3) | 372 (208, 677) |
| Reg1B (μg/mL), median (Q1, Q3) | 48.8 (13.9, 88.4) |
| LCN2 (μg/g), median (Q1, Q3) | 134 (41.6, 280) |

AGP, alpha-1-acid glycoprotein; CRP, C-reactive protein; LCN2, lipocalin-2; LRP1, low-density lipoprotein receptor-related protein-1; MPO, myeloperoxidase.

Reg1B and EED score. Correlations among different variables are illustrated in table 3.

## DISCUSSION

Our results reveal that LCN2 positively correlates with biomarkers of intestinal inflammation including MPO, NEO, calprotectin and Reg1B. These results are in accordance with the findings of another study, where LCN2 has been found to be positively correlated with MPO and calprotectin.[27] Another study also reported positive correlation between LCN2 and Reg1B, this is also in line with our findings.[28] Moreover, MPO, NEO, caprotectin have been emerged as a biomarkers of intestinal inflammation as well as predictor of EED.[26 29–32]

**Table 3** Correlation of LCN2 and LRP1 with other faecal biomarkers and EED score

| Variables | LRP1 | LCN2 |
|---|---|---|
| LRP1 | 1.00 | |
| LCN2 | −0.04 | 1.00 |
| MPO | −0.18* | 0.19* |
| NEO | −0.30* | 0.20* |
| A1AT | −0.18* | −0.005 |
| Calprotectin | −0.12 | 0.30* |
| Reg1B | −0.20* | 0.20* |
| EED score | −0.29* | 0.20* |

*Statistically significant (exact p value are shown in online supplemental file 1).
EED, environmental enteric dysfunction; LCN2, lipocalin-2; LRP1, low-density lipoprotein receptor-related protein-1; MPO, myeloperoxidase; NEO, neopterin.

Therefore, LCN2 might be a potential biomarker to diagnose intestinal inflammation and EED. Study also reported that LCN2 is released from various cell types and upregulated in tissue damaging conditions such as ulcerative colitis, infection and burn injury.[6 27] It has been conceded as an promising biomarker of inflammation, infection, ischaemia as well as kidney damage.[6] LCN2 is highly expressed in intestinal epithelial cell in case of inflammatory bowel disease.[6] Moreover, it has been reported as a dynamic and sensitive biomarker of gut inflammation.[33] Besides, LCN2 also positively correlated with EED score. In addition, higher expression of LCN2 gene was found in duodenal biopsies of EED in transcriptomic study.[34] Another study also reported that there was increased expression of LCN2 gene in case of severe acute malnutrition enteropathy.[35] Overall, our finding implies that LCN2 might be a potential biomarker for intestinal inflammation as well as it may be used as a predictor of EED.

In our analysis, we also measured the level of plasma LRP1. Correlation among the plasma LRP1 and faecal biomarkers showed that LRP1 is negatively correlated with MPO, NEO, A1AT and Reg1B. Moreover, LRP1 also negatively correlates with EED score. MPO, NEO, A1AT have been reported as biomarkers of intestinal inflammation and EED[26 29–31] and correlates negatively with LRP1. Therefore, these results support our supposition as we hypothesised that LRP1 may reduce intestinal inflammation. Overall, it can be inferred that LRP1 may contribute in the reduction of intestinal inflammation. Our finding is supported by another study, where it has been reported that LRP1 exerts anti-inflammatory effect on LPS exposure.[36] LRP1 also functions in activation of lysosomal enzymes, phagocytosis as well as destruction of micro-organisms thus LRP1 contributes in reduction of gut inflammation.[16 19 37] LRP1 was also found to be played a pivotal role in elimination of pathogen by scavenging membrane lipoproteins of microorganism during microbial invasion in another study.[23] On the other hand, intestinal inflammation has been observed in LRP1 knockout mice.[4] Altogether, these findings strongly suggest that LRP1 has an effect on the reduction of gut inflammation.

### Limitations and strengths

There were several limitations in our study. First, we could not measure the genetic expression of LCN2 and LRP1 of participants due to invasive collection process of intestinal cell. Second, we did not examine the consequence of suppression of LRP1 expression of human to investigate whether decreased expression may or may not contribute to gut inflammation because of ethical consideration. Lack of age-matched and sex-matched healthy children was also a limitation in this study. However, this was the first study that investigated the relationship of LCN2 and LRP1 with biomarkers of EED.

## CONCLUSIONS

Our findings suggest that LCN2 might be a potential biomarker to predict the gut inflammation and EED. However, increased level of LRP1 may contribute in alleviating intestinal inflammation that could be a potential therapeutic target to reduce the inflammation.

**Author affiliations**
¹Nutrition and Clinical Services Division (NCSD), International Centre for Diarrhoeal Disease Research Bangladesh (icddr,b), Dhaka, Bangladesh
²Department of Global Health, University of Washington, Seattle, Washington, USA

**Acknowledgements** We cordially thank all the participants and their mothers of the BEED study as well as the laboratory and field staffs at icddr,b for their contributions. icddr,b also gratefully acknowledges the donors (Bill & Melinda Gates Foundation) of this research protocol.

**Contributors** TA conceptualised the study. MMH wrote the manuscript and analyzed the data. MMH, MAG and FH did the laboratory analysis. MAG, SD and SMF critically reviewed the manuscript. MAA and MMH involved in statistical analysis. MM and TA involved in the development of study protocol. All authors commented on manuscript and approved the final version.

**Funding** This study was funded by the Bill & Melinda Gates Foundation. The investment ID was OPP1136751.(https://www.gatesfoundation.org/How-We-Work/Quick-inks/GrantsDatabase/Grants/2015/11/OPP1136751).

**Competing interests** No, there are no competing interests.

**Patient consent for publication** Not required.

**Ethics approval** The study protocol was approved by Institutional review board of the International Centre for Diarrhoeal Disease Research, Bangladesh (icddr,b). Protocol number-PR-16007.

**Provenance and peer review** Not commissioned; externally peer reviewed.

**Data availability statement** No data are available. The data set that was created during the study is not publicly available due to the restriction of funder. However, suggestion for data analysis can be made to corresponding author.

**ORCID iDs**
Md. Amran Gazi http://orcid.org/0000-0002-3286-7536
Tahmeed Ahmed http://orcid.org/0000-0002-4607-7439

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
