## [Reviewer comments · BMJ Paediatrics Open]

ARTICLE DETAILS

TITLE (PROVISIONAL)	Association of Lipocalin-2 and Low-density lipoprotein receptor related protein-1 (LRP1) with biomarkers of Environmental Enteric Dysfunction (EED) among under 2 children in Bangladesh: Results from a community based intervention study
AUTHORS	Gazi, Md Amran Hasan, Md. Mehedi Das, Subhasish Fahim, Shah Mohammad Hossaini, Farzana Alam, Md. Ashraf; Mahfuz, Mustafa Ahmed, Tahmeed

VERSION 1 – REVIEW

REVIEWER	Reviewer name: Dr. Peter Flom Institution and Country: Peter Flom Consulting, United States Competing interests: None
REVIEW RETURNED	27-Apr-2021

GENERAL COMMENTS	I confine my remarks to statistical aspects of this paper. Unfortunately, I think the chosen analysis is incorrect. First, the children should not be grouped into "responded" and "failed to respond". Surely some responded very well, some only a little, some not at all, and (perhaps) some got worse. Categorizing the dependent variable lowers power, and increases type I error. Second, since the key thing seems to be change in the dependent variables and, since the children seem to have been measured more than once, the errors from any analysis will be dependent and this violates the assumptions of all the tests done. Dependent errors can be accounted for by either multilevel models or generalized estimating equations. Either is OK, but I suggest MLM. Finally, the method of variable selection is not correct. It is known as bivariate screening and can be shown to give incorrect output (p values are too low, standard errors are too small, parameter estimates are biased away from 0, etc.) for more on this, see Harrell *Regression Modeling Strategies*. It is better to use substantive knowledge for model building, but if the authors insist on an automatic method, LASSO is better than most other methods. Peter Flom
--

REVIEWER	Reviewer name: Dr. Kanta Chandwe Institution and Country: University of Zambia School of Medicine
-----------------	--

	Paediatrics and Child Health, Zambia Competing interests: none
REVIEW RETURNED	17-May-2021

GENERAL COMMENTS	Thank you for the opportunity to review this manuscript looking at a very important topic that affects many children from LMICs. Non-invasive biomarkers of EED will definitely go a long way in identifying affected children and possibly treatment options. The following are my comments and questions for the authors: (1) In the methods section, line 138 states that the 222 participants had LAZ <-1. Please elaborate how many children were malnourished (LAZ and or WLZ <-2SD, or MUAC <12.5cm) at baseline. (2) Did all the children receive nutritional rehabilitation? if so, what was the justification of giving this to children who were not stunted (LAZ<-2SD) or wasted (WLZ <-2SD)? (3) Table 1 only shows characteristics after the intervention, What were the baseline characteristics before interventions? This will help compare the two groups better . The result section will benefit from a baseline characteristics table. (4) Table 2: This comparison is between stunted (mean LAZ -2.21) and not stunted (mean LAZ -1.37) children. Line 199 states "The levels of plasma biomarkers including LRP1, AGP, ferritin and fecal biomarkers including NEO, 200 AAT, Reg1B, and LCN2 were higher in nutritionally improved participants". Apart from sCD14 and LCN2, there are no statistically significant differences between the groups. (5) Table 2: What were the baseline plasma and fecal biomarkers? (6) Discussion: The discussion mainly repetition of results and not focused on discussing the findings. Please elaborate on what the results mean in the context of biomarkers and EED (7) Discussion: the first paragraph discusses LCN2 as a potential biomarker for gut inflammation and thus for EED. However, LCN2 was significantly higher in children who improved (not malnourished) as opposed to being higher in stunted children in whom EED and thus gut inflammation is likely to be more severe. How do the authors reconcile this? Thank you.
--

VERSION 1 – AUTHOR RESPONSE

BMJ Paediatrics Open

Dear Editor and Reviewers,

Thank you very much for reviewing the manuscript entitled "Association of Lipocalin-2 and Low-density lipoprotein receptor related protein-1 (LRP1) with biomarkers of Environmental Enteric Dysfunction (EED) among under 2 children in Bangladesh: Results from a community based intervention study". Please find below the responses to the comments made by the editor and reviewers and a description of the changes made in the manuscript

Associate Editor

Comments to the Author:

This is interesting work on an important topic, however both the reviewers raise important points that should be addressed. The statistical concerns in particular are significant enough that I think a complete reanalysis is needed before publication could be considered.

Response: Thank you for your overall positive feedback. We have addressed the comments point-by-point raised by the editor and reviewers.

I have a few additional comments:

Comments: The categorization of subjects into improved/not improved for purposes of looking at the biomarkers is not a very useful classification, since this incorporates too great a range of what might or might not be clinically or biologically significant change.

Response: Thank you for your suggestion. We have omitted the categorization of subject into improved/not improved according your suggestion. Now, we have just reported as "Baseline" and "End line" data point (Table-1 and Table-2).

Comments: In addition, as Jef Leroy and others have shown fairly convincingly there is a great deal of uncertainty about whether serial Z scores can be followed in this fashion (without a control group to permit change/difference comparisons)

(see:

https://linkprotect.cudasvc.com/url?a=https%3a%2f%2fbmcpediatr.biomedcentral.com%2farticles%2f10.1186%2fs12887-015-0458-9&c=E,1,ejP1nC-_j4v-6H8TMPklcJNSF3rlyk1cyV8m-d_hrJW97TT6uBi0-cNMNQ1EUL57RunQV5dMyfsazxm7DcxeiZ5gSVRkMdHnJH_q3PqxqeVErIJw,,&typo=1 and other publications). This is because the variance of the WHO standard population increases with age, so at a constant height deficit as the child ages the Z score will decrease. At a minimum within subjects modelling is needed (as the stats reviewer suggests) and building statistical models based on the continuous growth variable. It would also be best to see a look at height-age-difference in addition to Z scores.

Response: Thank you for your observation. This is a sub study of Bangladesh environmental enteric dysfunction (BEED) study: ClinicalTrials.gov ID: NCT02812615. According to the objectives of BEED study, LAZ score has been calculated as a growth parameter. Unfortunately, data have not been collected to calculate the "height for age differences (HAD)" in this study. But, your suggestion is very important to us, hope we will be able to conduct a similar type of study considering HAD.

Comments: The timing of the biomarkers vs the growth surveillance is not clear. These details should be provided. For example, do "improved" vs "not improved" mean a baseline/endline growth data point?

Response: Thank you for your comment. "Improved" vs "not improved" means the "end line" growth data point. In our study, "improvement" has been defined by "any positive change of LAZ" at the end line from baseline after receiving nutritional intervention. However, "no change of LAZ" at end line from baseline has been considered as "non-improvement". That was mentioned in the "main document" (Line-140-143). According to the objective of the BEED study, firstly participants were enrolled and their "baseline" data were collected during enrolment and these participants were given nutritional intervention (The intervention package included an egg, 150 ml of milk, micronutrient powder and nutritional counseling) for 90 days, this intervention starts immediately after baseline data collection. After completion of the intervention the "endline" data were collected from the participants (Line-138-142).

Comments: Are their multiple data points for several different months, and over what time (if multiple, then where does the improved/not improved metric come from?).

Response: Thank you for your question. Data were collected only at two-time points (Baseline and endline data point). The time between baseline and endline is 90 days.

Comments: In the published study protocol, biomarkers were drawn before and after the intervention. Here, it isn't clear if analysis is based on the before or after values, and why not change in value is not

analyzed as appears to be the case for the growth data.

Response: Thank you for your concern. For this particular analysis (correlation and regression), we have only considered and included the data that were taken after nutritional intervention (end line) (Line: 182).

Comments: The correlations between biomarkers and EED scores seem the most relevant, whereas the within-biomarkers correlations are less interesting and difficult to interpret. I would focus on just the LRP1 and LCN2 correlations.

Response: Thank you for your suggestion. We have addressed accordingly. We have omitted the correlation between other biomarkers from table-3.

Comments: Along this line, the stats reviewer has concerns about regression models not being built on substantive knowledge in favor of p value screening. These leads to models that are also hard to interpret. For example, in the model on growth improvement (Table 4) it seems like the independent variable of interest here would be the EED score and potentially LCN2 and LRP1 (although there is almost certain interaction between the EED score and the biomarkers) as opposed to other biomarkers (Neopterin) which aren't the subject of investigation here. Similarly low ferritin as a marker for growth faltering is surprising as it is often a systemic marker of inflammation, and this makes me wonder about imbalance in the groups (which could be sorted with a multilevel model).

Response: Thank you for your concern. As per your suggestion in the first comment, we have avoided the categorization into improved /not-improved. Therefore, table-4 has been omitted as it has been analyzed on the basis of improved and not-improved categories. Moreover, we built the logistic model (Table-4) to incorporated all the selected variables on the basis of literature review (Detailed literature review has been discussed in the response against second comment of the first reviewer). In addition, we agree with you that ferritin is a marker of systemic inflammation but study also reported that there was a significant correlation between serum ferritin level and growth disorders [1]. We also incorporated the Neopterin in multivariable logistic model (Table-4) on the basis of literature review [2], where it has been found to be negatively associated with linear growth.

Comments: The appendix with its regression models is of limited interest, since there is no justification for why this detailed exploration of some but not other biomarker relationships is undertaken. It would be much more compelled to focus on the novel items under consideration here (LCN2 LRP1) and clinical endpoints (EED score, growth).

Response: Thank you for your suggestion. We have omitted the Table-S1, S2 and S3 from the appendix.

Reviewer: 1

Dr. Peter Flom, Peter Flom Consulting

Comments to the Author

I confine my remarks to statistical aspects of this paper. Unfortunately, I think the chosen analysis is incorrect.

Comments: First, the children should not be grouped into "responded" and "failed to respond". Surely some responded very well, some only a little, some not at all, and (perhaps) some got worse. Categorizing the dependent variable lowers power, and increases type I error.

Response: Thank you for your suggestion. We have omitted the categorization of children into improved/not improved according your suggestion. Now, we have just reported as "Baselin" and "End line" data (Table-1 and Table-2).

Comments: Second, since the key thing seems to be change in the dependent variables and, since the children seem to have been measured more than once, the errors from any analysis will be dependent and this violates the assumptions of all the tests done. Dependent errors can be accounted for by either multilevel models or generalized estimating equations. Either is OK, but I suggest MLM.

Response: Thank you for your suggestion. For this particular analysis (Correlation and regression), we have only considered and included the data that were taken after nutritional intervention (end line).

Comments: Finally, the method of variable selection is not correct. It is known as bivariate screening and can be shown to give incorrect output (p values are too low, standard errors are too small, parameter estimates are biased away from 0, etc.) for more on this, see Harrell *Regression Modeling Strategies*. It is better to use substantive knowledge for model building, but if the authors insist on an automatic method, LASSO is better than most other methods.

Response: Thank you for your suggestion. As you and associate editor, suggested to avoid the categorization into improved /not-improved in first comment, so we have omitted the table-4 as because this has been analyzed on the basis of improved and not-improved categories. Moreover, we have selected the variables including age [3], sex [4, 5], maternal height [6], family income [7], ferritin [1, 8, 9], sCD14 [10], neopterin [2] to incorporate into the multivariable logistic model on the basis of literature review and these variables were found to be associated (positively or negatively) with linear growth in different studies. Moreover LCN2 has been incorporated as covariate because it has been reported as growth factor, differentiation factor and adipokine in various studies [11-14].

Peter Flom

Reviewer: 2

Dr. Kanta Chandwe, University of Zambia School of Medicine **Comments to the Author** Thank you for the opportunity to review this manuscript looking at a very important topic that affects many children from LMICs. Non-invasive biomarkers of EED will definitely go a long way in identifying affected children and possibly treatment options.

The following are my comments and questions for the authors:

(1) In the methods section, line 138 states that the 222 participants had LAZ < -1. Please elaborate how many children were malnourished (LAZ and or WLZ < -2SD, or MUAC < 12.5cm) at baseline.

Response: Thank you for your query. Among the participants, 115 were stunted (LAZ < -2 SD) at the base line.

(2) Did all the children receive nutritional rehabilitation? if so, what was the justification of giving this to children who were not stunted (LAZ < -2SD) or wasted (WLZ < -2SD)?

Response: Thank you for your comments. Yes, all the enrolled children received nutritional intervention. This is a sub-study of Bangladesh environmental enteric dysfunction (BEED) study: ClinicalTrials.gov ID: NCT02812615. This was the aim of BEED study to give nutritional intervention to all the children (stunted and not stunted). Moreover, "not stunted" children were considered as "at risk of being stunted (LAZ: -1 to -2)" children in BEED study.

(3) Table 1 only shows characteristics after the intervention, what were the baseline characteristics before interventions? This will help compare the two groups better. The result section will benefit from a baseline characteristics table.

Response: Thank you for your suggestion. We have addressed accordingly (Table-1).

(4) Table 2: This comparison is between stunted (mean LAZ -2.21) and not stunted (mean LAZ -1.37) children. Line 199 states "The levels of plasma biomarkers including LRP1, AGP, ferritin and fecal biomarkers including NEO, Line-200 AAT, Reg1B, and LCN2 were higher in nutritionally improved participants". Apart from sCD14 and LCN2, there are no statistically significant differences between the groups.

Response: Thank you for your concern. Yes, apart from the sCD14 and LCN2, significant differences were not found among other biomarkers. As suggested by reviewer-1 and associate editor, we have avoided the categorization into improved /not-improved, so we have also omitted the categorization into improved/not-improved in table-1 & 2.

(5) Table 2: What were the baseline plasma and fecal biomarkers?

Response: Thank you for your concern. We have addressed accordingly (Table-2).

(6) Discussion: The discussion mainly repetition of results and not focused on discussing the findings. Please elaborate on what the results mean in the context of biomarkers and EED

Response: Thank you for your suggestion. We have addressed according your suggestion, that we have omitted the discussion line: 250- 269 on growth parameter because we have deleted the table-4. According to the title of the manuscript, the novel biomarkers are LCN2 and LRP1, for this reason we only focus on the correlation of these two biomarkers with available biomarkers (MPO, Neopterin, AAT, Cal, Reg1B) of gut inflammation and EED. Moreover, we have deleted the line-242 to overcome the repetition. We have also included some supported documents about LCN2 and EED in discussion section (Line: 248-251).

(7) Discussion: the first paragraph discusses LCN2 as a potential biomarker for gut inflammation and thus for EED. However, LCN2 was significantly higher in children who improved (not malnourished) as opposed to being higher in stunted children in whom EED and thus gut inflammation is likely to be more severe. How do the authors reconcile this?

Response: Thank you for your comment. Yes, LCN2 was significantly higher in improved participant, this might because improved participants were still undernourished (mean LAZ: -1.37) after nutritional intervention. Though LCN2 has been reported as biomarker of inflammation, it also reported as growth factor, differentiation factor and adipokine in various studies [11-14], that's why it may have capacity to improve the health condition, but, larger experimental trail is warranted to find the mechanism by which LCN2 shows its health improving role. Moreover, we have omitted the categorization into improved/not-improved in table-1 & 2.

Thank you.

References:

1. Rathaur, V.K., A. Imran, and M. Pathania, Growth pattern in thalassemic children and their correlation with serum ferritin. *Journal of family medicine and primary care*, 2020. 9(2): p. 1166.
2. Campbell, D.I., et al., Intestinal inflammation measured by fecal neopterin in Gambian children with enteropathy: association with growth failure, *Giardia lamblia*, and intestinal permeability. *Journal of pediatric gastroenterology and nutrition*, 2004. 39(2): p. 153-157.
3. Aguayo, V.M., et al., Determinants of stunting and poor linear growth in children under 2 years of age in India: An in-depth analysis of Maharashtra's comprehensive nutrition survey. *Maternal & child nutrition*, 2016. 12: p. 121-140.
4. Pryer, J.A. and S. Rogers, Epidemiology of undernutrition in adults in Dhaka slum households, Bangladesh. *European journal of clinical nutrition*, 2006. 60(7): p. 815-822.
5. Moestue, H., et al., Conclusions about differences in linear growth between Bangladeshi boys and girls depend on the growth reference used. *European journal of clinical nutrition*, 2004. 58(5): p. 725-731.
6. Addo, O.Y., et al., Maternal height and child growth patterns. *The Journal of pediatrics*, 2013. 163(2): p. 549-554. e1.
7. Sultana, P., M.M. Rahman, and J. Akter, Correlates of stunting among under-five children in Bangladesh: a multilevel approach. *BMC nutrition*, 2019. 5(1): p. 1-12.
8. Peng, W., et al., Iron status and linear growth: a prospective study in school-age children. *European journal of clinical nutrition*, 2013. 67(6): p. 646-651.
9. Soliman, A.T., et al., Linear growth in children with iron deficiency anemia before and after treatment. *Journal of tropical pediatrics*, 2009. 55(5): p. 324-327.
10. Zambruni, M., et al., Stunting is preceded by intestinal mucosal damage and microbiome changes and is associated with systemic inflammation in a cohort of Peruvian infants. *The American journal of tropical medicine and hygiene*, 2019. 101(5): p. 1009-1017.
11. Chakraborty, S., et al., The multifaceted roles of neutrophil gelatinase associated lipocalin (NGAL) in inflammation and cancer. *Biochimica et Biophysica Acta (BBA)-Reviews on Cancer*, 2012. 1826(1): p. 129-169.

12. Devireddy, L.R., et al., A cell-surface receptor for lipocalin 24p3 selectively mediates apoptosis and iron uptake. *Cell*, 2005. 123(7): p. 1293-1305.
13. Wang, Y., et al., Lipocalin-2 is an inflammatory marker closely associated with obesity, insulin resistance, and hyperglycemia in humans. *Clinical chemistry*, 2007. 53(1): p. 34-41.
14. Yan, Q.-W., et al., The adipokine lipocalin 2 is regulated by obesity and promotes insulin resistance. *Diabetes*, 2007. 56(10): p. 2533-2540.

VERSION 2 – REVIEW

REVIEWER	Reviewer name: Dr. Peter Flom Institution and Country: Peter Flom Consulting, United States Competing interests: None
REVIEW RETURNED	01-Jul-2021

GENERAL COMMENTS	The authors have addressed my concerns and I now recommend publication. Peter Flom
---

VERSION 2 – AUTHOR RESPONSE

BMJ Paediatrics Open

Dear Editor and Reviewers,

Thank you very much for reviewing the manuscript entitled "Association of Lipocalin-2 and Low-density lipoprotein receptor related protein-1 (LRP1) with biomarkers of Environmental Enteric Dysfunction (EED) among under 2 children in Bangladesh: Results from a community based intervention study". Please find below the responses to the comments made by the editor and a description of the changes made in the manuscript.

Editor in Chief Comments to Author:

Comments: What is already known replace "developing countries" with "low and middle income countries". Introduction 1st sentence replace "developing countries" with "low and middle income countries"

Response: Thank you for your suggestion. We have addressed accordingly (Line: 66 & 100).

Associate Editor

Comments to the Author:

Comment: This is much clearer now and a nice contribution to the biomarker literature.

Response: Thank you so much for your positive feedback and considering our manuscript.

Comments: It is a little unfortunate that the authors don't take up the stats reviewer's suggestion to use mixed models to include pre and post values with random effects for individuals, as this would have been the way to use the data to their maximum. However, a single time point correlation analysis is acceptable and interesting I think.

Response: Thank you for your observation. For this particular analysis, we have used the data from BEED study, a larger community based intervention study. The detail methodology of BEED has been published elsewhere [1]. In the current analysis, our objective was to investigate the two new biomarkers such as LCN2 and LRP1 as biomarkers of EED in children. However, there was no baseline data for LCN2 in BEED study, as these were initially not included in the BEED methodology. Therefore, we could not able to include the stats reviewer's suggestion and not able to made the necessary correction. However, we appreciate your interest on single time point analysis and think that this would be interesting to the readers.

Comments: However, the choice to use only endline data in the analysis raises the question as to why the baseline data is included at all since it is not used for anything. I think the authors could shorten the discussion of the intervention (since change values are no longer being considered) and just present this as a cross sectional analysis, keeping only the time point that is included in the analysis.

Response: Thank you for your valuable suggestion. We have addressed accordingly (deleted the line: 135 – 139) to shorten the discussion of the intervention and also omitted the baseline data from Table-1 & 2. However, data for sex, maternal height, family income, improved sanitation and WAMI were only collected at baseline in BEED study. We believe that the data for these variables will be the same at end line. So we want to include these in the Table-1 with other variables those were actually collected at the end line.

Reviewer: 1

Dr. Peter Flom, Peter Flom Consulting

Comments to the Author

Comment: The authors have addressed my concerns and I now recommend publication.

Response: Thank you so much for your positive feedback.

Peter Flom

Reference:

1) Mahfuz M, Das S, Mazumder RN, et al. Bangladesh environmental enteric dysfunction (BEED) study: protocol for a community-based intervention study to validate non-invasive bio